# Illumination Adaptive Multi-Scale Water Surface Object Detection with Intrinsic Decomposition Augmentation

**Zhiguo Zhou \***, **Zeming Li †, Jiaen Sun †, Limei Xu and Xuehua Zhou**

School of Integrated Circuits and Electronics, Beijing Institute of Technology, Beijing 100081, China; 3220210823@bit.edu.cn (Z.L.); 3120190696@bit.edu.cn (J.S.); 3220221541@bit.edu.cn (L.X.); xuehuazhou@bit.edu.cn (X.Z.)

**\*** Correspondence: zhiguozhou@bit.edu.cn; Tel.: +86-1368-334-5830

† These authors contributed equally to this work.

**Abstract:** Visual object detection is an essential task for the intelligent navigation of an Unmanned Surface Vehicle (USV), which can sense the obstacles while navigating. However, the harsh illumination conditions and large scale variation of the objects significantly harm the performance of object detection methods. To address the above problems, we propose a robust water surface object detection method named multi-scale feature fusion network with intrinsic decomposition generative adversarial network data augmentation (MFFDet-IDGAN). We introduce intrinsic decomposition as data augmentation for the object detection to achieve illumination adapting. And an intrinsic decomposition generative adversarial network (IDGAN) is proposed to achieve unsupervised intrinsic decomposition. Moreover, the multi-scale feature fusion network (MFFDet) adopts an improved bidirectional feature pyramid network (BiFPN) and spatial pyramid pooling (SPP) blocks to fuse features of different resolution for better multi-scale detection. And an improved weighted stochastic weight averaging (SWA) is proposed and applied in the training process to improve the generalization performance. We conduct extensive experiments on the Water Surface Object Detection Dataset (WSODD), and the results show that the proposed method can achieve 44% improvement over the baseline. And we further test our method on a real USV in the sailing process, the results show that our method can exceeding the baseline by 4.5%.

**Keywords:** multi-scale; intrinsic decomposition; water surface; object detection; generative adversarial network

## 1. Introduction

Unmanned surface vehicle (USV) has been widely applied in many fields, including military tasks, river topographic mapping, water quality monitoring and so on, for its minisize, lowcost and convenience. Visual perception is an important module for USV, for it can provide abundant information about the obstacles including location, speed counts and density. Object detection plays a major role in the relative image processing tasks, which can provide the category and pixel-level position of the objects. However, for the difference between the condition on the water surface and that on the ground, effective object detection for USV perception remains a challenging task.

### 1.1. Background

To achieve automatic navigation, object detection is a key module for USV, for it can sense and locate the obstacles around the vehicle. With the great success of convolutional neural networks and deep learning in the computer vision community [1,2], many effective object detection methods have been proposed. YOLO [3–6] and SSD [7] are widely applied object detection algorithms for their fine accuracy and rapidness. However, these methods are designed and tested in general scenes, while they obtain poor performance in the water surface scenes. The property of the objects and scenes on the water surface are greatly

different from that in general scenes, so there exists several challenges to design an effective object detection method.

One of the challenges is that the range of distance variation from the target is large, so the scale of the objects varies greatly. The objects of extreme scale (extremely large or extremely small) can be more likely to cause missed detection. In the field of water surface object detection, many studies have been carried out to realize multi-scale detection. Most of these works achieve multi-scale detection by improving the feature fusion strategy [8–11], while others utilize data augmentation [12] and scene narrowing [13].

Another challenge is that the illumination condition on the water surface can be complex and harsh, for there exist far more specular reflections. The strong illumination can badly affect the quality of image, resulting in difficulty in distinguishing the features of the targets. Additionally, because there is no additional light source, the illumination condition greatly depends on the sunlight, which results in massive backlight scenes. In the backlight scenes, we will face a condition that the illumination on the object is extremely poor while extremely strong light in the background, both of which would harm the performance of the object detection. However, previous works hardly focus on this issue. Some methods choose to improve the illumination condition by preprocessing the input image. Zhang et al. [14] utilize gamma correction as a preprocess method to reduce the interference of sunlight and Syed [15] makes use of a specular removal method to reduce the reflection on the water surface. Preprocessing image to improve the illumination condition is a seemingly feasible method, and there are many works for specular removal by digital image processing [16–19] and deep learning based methods [20–22]. However, the digital image processing methods can hardly deal with the complex illumination conditions on water surface, these methods can cause severe image color distortion, which may decrease the performance of detection method in contrary. The deep learning methods can be more powerful for its strong feature extraction ability, but the large calculation of deep neural network can cause a significant increase in the calculating time, which is not acceptable for USVs with limited computing resources.

Based on the above analysis, to achieve illumination adapting and multi-scale detection for water surface scenes, a multi-scale feature fusion object detection with intrinsic decomposition generative adversarial network data augmentation (MMFDet-IDGAN) is proposed. And to enhance the generalization ability of the model, an improved Weighted-SWA is proposed and applied. Extensive experiments on WSODD show that the proposed method can achieve 44% improvement over the baseline. And the experiments on real world sailing also demonstrate that our method can obtain better accuracy than other methods with equal rapidness.

### 1.2. Contributions

The contributions of this paper include:

1. To deal with the harsh and complex illumination condition on water surface scenes, we introduce intrinsic decomposition as a data augmentation method to enable the object detection network to adapt to the harsh illumination condition on water surface scenes. And the results of experiments demonstrate that it is an effective way to handle the complex illumination condition without any extra calculation while detecting. For the lack of high quality annotated intrinsic decomposition datasets, we propose an unsupervised method named intrinsic decomposition generative adversarial network (IDGAN) to address this task. The natural images in the dataset are decomposed to reflectance and shading to obtain more prior information to achieve illumination adapting.

2. To obtain better performance while detecting the objects with extreme scale, we proposed a multi-scale feature fusion object detection network (MFFDet) to improve the multi-scale detection effect. The network take use of a deeper CSPDarknet53 to obtain more effective semantic features. And a multi-scale feature fusion neck with

spatial pyramid pooling (SPP) blocks and improved bidirectional feature pyramid network (BiFPN) is used to improve the multi-scale detection performance.

3.  To obtain a model with better generalization, an improved model ensembling method Weighted-SWA is proposed, which utilizes entropy evaluation to weight the models to ensure that the models converge to the optimal solution region. The Weighted-SWA can enhance the generalization of the model by ensuring that the model is located in the smooth region of the solution space.

### 1.3. Organization

The remainder of this paper is structured as follows. Section 2 reviews the previous approaches related to water surface object detection and intrinsic decomposition. Section 3 describes the proposed method in detail. Section 4 presents the experimental results of our method in dataset experiments and practical verification. In Section 5, we analyze the experiment results and potential challenges. And we finally give a summarize of our work in Section 6.

## 2. Related Works

### 2.1. General Object Detection Methods

Existing object detection methods mainly include anchor-based methods, anchor-free methods, and transformer methods. Anchor-based methods can be further divided into two-stage methods and one-stage methods.

Anchor-based methods set several prior boxes to obtain further classification and regression. Two-stage methods (R-CNN [23], SPPNet [24], Fast R-CNN [25], Faster R-CNN [26], Mask R-CNN [27], and Cascade R-CNN [28]) apply region proposal networks (RPN) to obtain a region of interest to obtain further classification and regression. The two-stage methods can be more precise but less efficient due to the complex process. One-stage (YOLOv1-v4 [3–6], SSD [7], RetinaNet [29]) methods generate classification and regression results directly from the prior boxes. Without the RPN and box reinforcement, one-stage methods can be faster but not so accurate as the two-stage ones. For the effectiveness and rapidness of one-stage methods, they are widely applied in engineering conditions. The use of anchor enable the designer to add prior knowledge to increase the stability and robustness of the network, while the anchors cost much computation resource. And there exists the problem of imbalance of positive and negative samples with the use of an anchor, resulting in a decrease in the accuracy.

For the drawbacks carried by the usage of an anchor, the anchor-free methods (FCOS [30], CornerNet [31], CenterNet [32]) are widely researched to achieve better detection effects. FCOS uses a full convolutional network to detect objects, which is similar to semantic segmentation. CornetNet locates the object by its top left corner and bottom right corner. Centernet represents the object as a Gaussian circle in the heatmap. The anchor-free methods can obtain faster detection without anchors, but they can hardly obtain better precision than the anchor-based ones.

Due to the great success of the transformer model in neural language processing, it also draws much attention in the computer vision field. DETR [33] is the first one using transformer to implement object detection tasks. It directly generates the class and location of objects by a transformer encoder and decoder, which can achieve end-to-end object detection. Inspired by DETR, many transformer-based object detection methods have been proposed and obtained promising performance [34,35].

These methods are train and tested on the datasets of general scenes, which can be badly affected by some noise such as extremely illumination. So, some improvements should be applied to obtain better detection performance on special scenes.

### 2.2. Object Detection Methods for Water Surface Scenes

To sense the obstacles on the water surface, many sensing methods are proposed. But many research about object detection on water surfaces are based on infrared or radar

images, while few works are proposed for the object detection on water surfaces using visual images.

An et al. [36] proposed a modified water surface object detection backbone based on rbox, to obtain the recall rate and precision. Li et al. [37] proposed a water surface object detection method Yolov3-2SMA, which can achieve real-time object detection with high precision in a dynamic water environment. Zhang Sr et al. [38] used a modified YOLO and multi-feature detecting method to detect ships. It applies multi-dimensional scaling (MDS) for dimensionality reduction of SIFT features, and uses random sample consensus to optimize SIFT matching, which reduces miss-matching effectively. Jie et al. [39] modified YoloV3 for detecting ships on inland rivers, which increases the mAP and FPS by 5% and 2%, respectively.

For the importance of effective visual object detection and the lack of studies on water surface object detection, it is essential to conduct more research on water surface object detection to obtain better sensing effects.

*2.3. Intrinsic Image Decomposition*

The concept of intrinsic decomposition originated from Retinex theory [40], and was first proposed by Barrow and Tenenbaum [41]. Intrinsic decomposition aims to decompose a natural image into an illumination-variant component and illumination-invariant component, so it can benefit many high-level computer vision tasks.

Intrinsic image decomposition has been researched for nearly fifty years, and the research can be divided into non-learning-based methods and learning-based methods.

The non-learning-based methods tend to solve a non-learning optimization with handmade prior constraints. Many methods set prior constraints based on empirical observation to extract the reflectance information of natural images [42,43]. These methods establish a connection between reflectance and the chromaticities of pixels to inference reflectance images and further obtain shading components. Shen et al. [44] propose to add texture cues to the traditional methods to obtain better reflectance results. Zhao et al. [45] formulate intrinsic decomposition as the minimization of a quadratic function, which incorporates both the Retinex constraint and our nonlocal texture constraint.

In recent years, many learning-based methods have been proposed, which include supervised and unsupervised methods. The supervised methods tend to directly learn the distribution of reflectance and shading from labeled reflectance and shading training samples [46–48]. But most of the public intrinsic decompositon datasets are highly synthetic [47,49,50], for which the methods trained on them cannot performance well in the real-world scenes. And other datasets consisting of real images are too small to support supervised methods to obtain competitive generalization performance [51]. There also exist several unsupervised methods. Janner et al. [52] use unsupervised reconstruction error as an additional signal to make use of the plentiful unlabeled data. Zhang et al. [53] proposed an unsupervised intrinsic decomposition method based on the observations that the reflectance of a natural image typically has high internal self-similarity of patches. Lettry et al. [54] build an unsupervised intrinsic decomposition method based on the analysis of albedo that is invariant to lighting conditions, and cross-combining the estimated albedo of a first image with the estimated shading of a second one should lead back to the second one's input image.

As an effective unsupervised method, generative adversarial network [55] achieved great success in many tasks, so it is a promising method to achieve effective unsupervised intrinsic decomposition. But few studies are conducted on intrinsic decomposition generative adversarial network, for which it' is essential to apply more research on to explore the effectiveness.

## 3. Method

### 3.1. Overall Architecture

The images capture by a sailing USV are different from the generally captured ones, especially the large scale variance and harsh illumination condition. To support effective perception for USV, we construct MFFDet-IDGAN to achieve accurate and robust water surface object detection, as shown in Figure 1.

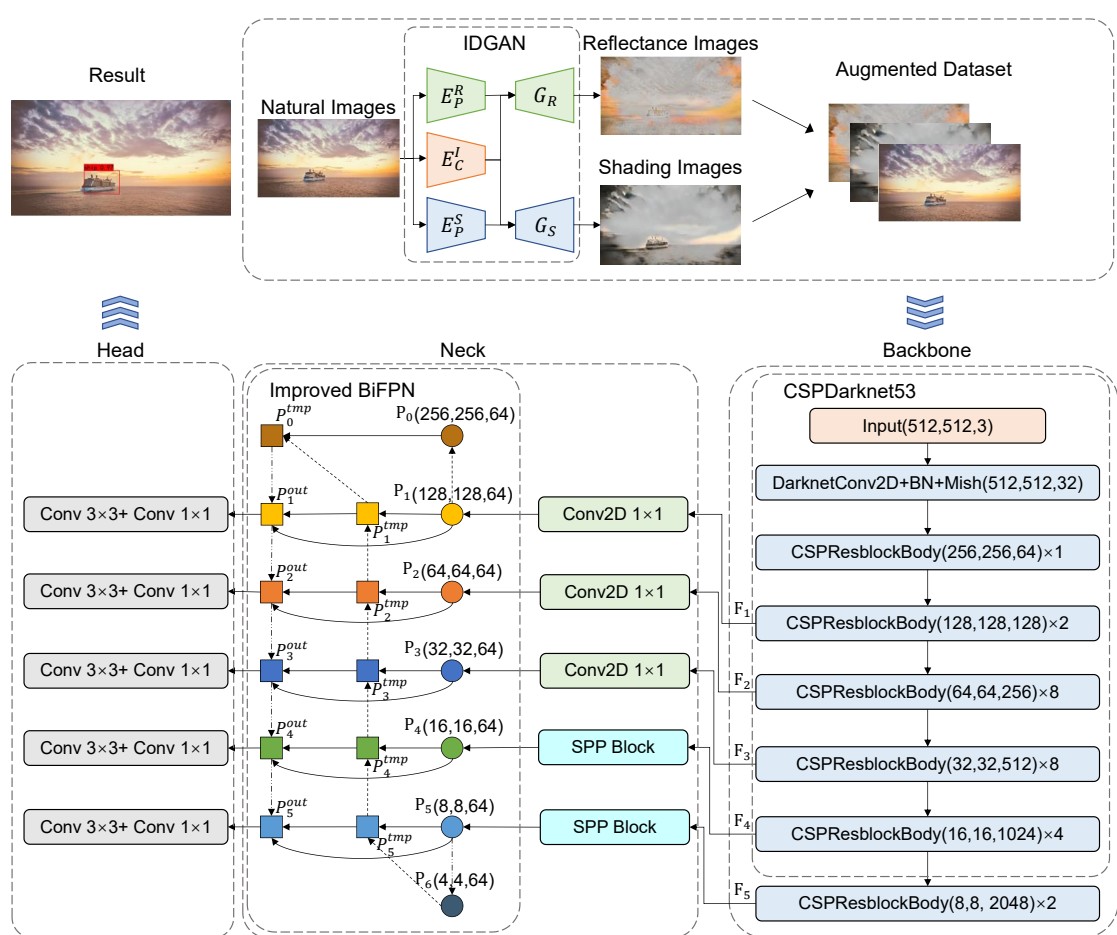

**Figure 1.** Overall architecture of MFFDet-IDGAN. The IDGAN (**upper part**) is only enabled in the training process for data augmentation. In the detection process, only MFFDet (**lower part**) is enabled.

Our method is composed of two parts. One part is intrinsic decomposition data augmentation (IDGAN), aiming to enrich the training set with decomposed images. Different from normal data augmentation methods (Cutmix [56], Mosaic [6], Random Affine and so on), intrinsic decomposition can revel the illumination and structural information of the images, which enable the object detection network to adapt to the harsh illumination condition. The other part is an multi-scale feature fusion object detection network (MFFDet), which utilizes improved modules to fuse features of different scales, aiming to achieve better multi-scale detection performance. Detailed description of our method is given in the following sections.

### 3.2. Intrinsic Decomposition Generative Adversarial Network

Retinex theory states that the information of an image is determined by the combined action of illumination and the structural property. Accordingly, a natural image can be decomposed to the reflectance image $R$ and shading image $S$, expressed as

$$I = R \times S \tag{1}$$

Equation (1) is ill-posed because the number of unknown variables is twice that of the known variables. To solve the problem, we utilize trainable modules to obtain the domain property of reflectance and shading, and further transfer the natural image to these domains with the learned knowledge. Recent image-style transfer methods assume that the content image shares the same content with the styled images. Furthermore, they assume that the content and the style information of an image can be separated by trainable encoders [57–59]. Similarly, intrinsic decomposition can be illustrated as a process that separates the content and domain information, then combine the content information with the reflectance and shading domain information. Accordingly, we regard intrinsic decomposition as a physically constrained image-style transfer task, and set several encoders to learn the domain prior knowledge to generate certain reflectance and shading images. Formally, the intrinsic decomposition task can be described as follows: given a set of images of different domains, including natural images $I$, reflectance images $R$ and shading images $S$ (the content in $I$, $R$ and $S$ can be totally different), we aim to learn domain transformation $I - R$ and $I - S$. To achieve this goal, we propose an intrinsic decomposition generative adversarial network (IDGAN), and the detailed implementation is illustrated in Figure 2.

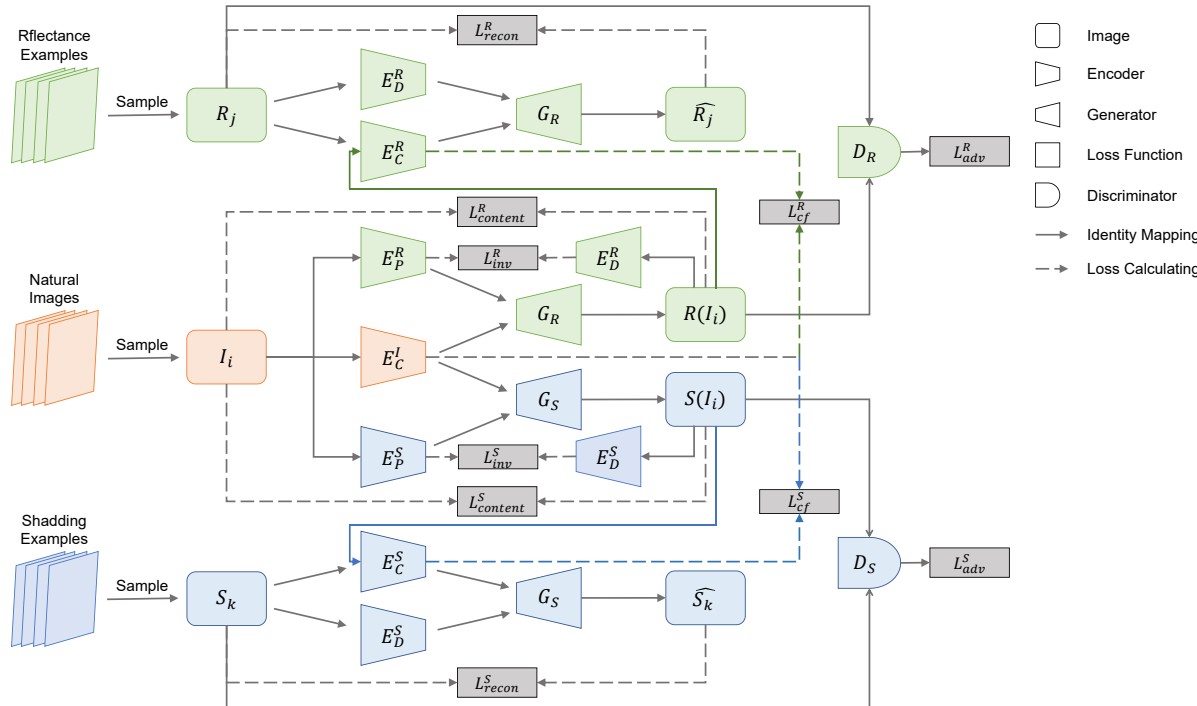

**Figure 2.** Architecture of IDGAN.

IDGAN consists of two branches: domain transformation branch and decoupling branch. The details of the two branches are illustrated as below.

**Domain transformation branch.** The domain transformation branch aims to transfer a natural image $I_i$ to reflectance image $R(I_i)$ and shading image $S(I_i)$. To this end, we utilize generators $G_R$ and $G_S$ to approximate the distribution of $R$ and $S$, by employing discriminators $D_R$ and $D_S$ to train against the generators. The joint training process of the generators and discriminators leads to generators that are able to produce images of the desired domains. Normal image transfer networks tend to take random noise as input to obtain more diverse-styled images, but from the perspective of intrinsic decomposition, we desire to obtain a definite output that is as close to the distribution of the desired domain as much. So we take the latent code of content and domain prior knowledge as input to feed the generators. To obtain the latent code of the two domains from the single input natural

image $I_i$, we utilize learnable prior knowledge encoders $E_P^R$ and $E_P^S$ to capture the domain code. And the adversarial losses are defined as follows:

$$\mathcal{L}_{adv}^R = log(1 - D_R(R(I_i))) + log(D_R(R_j)) \tag{2}$$

$$\mathcal{L}_{adv}^S = log(1 - D_S(S(I_i))) + log(D_S(S_j)) \tag{3}$$

The total adversarial loss is $\mathcal{L}_{adv} = \mathcal{L}_{adv}^R + \mathcal{L}_{adv}^S$

With only adversarial loss, the generators can produce images with the same distribution as the desired domains, but cannot keep the content information in $I_i$. To ensure that the generated images share the same content with the natural image, we utilize pixel-wise content loss $\mathcal{L}_{content}$ between the natural image $I_i$ and the domain transferred images $R(I_i)$ and $S(I_i)$. Moreover, to make the pixel-wise content loss not too strict, we apply an average pooling layer $P$ before calculating the loss. The pixel-wise content loss can be expressed as

$$\mathcal{L}_{content} = ||P(I_i) - P(S(I_i))||_2 + ||P(I_i) - P(R(I_i))||_2 \tag{4}$$

Furthermore, we desire the content features extracted by content encoders $E_C^I$, $E_C^R$, $E_C^S$ from $I_i$, $R(I_i)$, $S(I_i)$ be constant, so we set content feature loss $\mathcal{L}_{cf}$, expressed as

$$\mathcal{L}_{cf} = ||E_C^I(I_i) - E_C^R(R(I_i))||_2 + ||E_C^I(I_i) - E_C^S(S(I_i))||_2 \tag{5}$$

Equation (1) states that the natural image $I_i$ is equal to the pixel-product of the domain transferred images $R(I_i)$ and $S(I_i)$, so a physical constraint loss $\mathcal{L}_{phy}$ is employed to regularize our method, expressed as

$$\mathcal{L}_{phy} = ||I_i - R(I_i) \times S(I_i)||_2 \tag{6}$$

**Decoupling branch.** With the assumption that the content information and domain information of the image can be separated and the separated information can be used to reconstruct the image, we employ several content encoders $E_C$ and domain information encoders $E_D$, and set several losses to ensure the encoders can extract desired information.

To simplify the description, we will focus on the reflectance part, and the shading part is constructed in the same way. For a reflectance sample $R_j$, $E_C^R$ and $E_D^R$ are applied to obtain the content and domain latent codes, then reconstruct $\hat{R}_j$ by generator $G_R$. We utilize reconstruction loss $L_{recon}$ to constrain this process. The reconstruction loss is expressed as

$$\mathcal{L}_{recon} = ||\hat{R}_j - R_j||_2 + ||\hat{S}_k - S_k||_2 \tag{7}$$

With the reconstruction loss $\mathcal{L}_{recon}$ and the content feature loss $\mathcal{L}_{cf}$, we can obtain an effective domain information encoder $E_D^R$ to access the domain information of a image. To ensure the certainty of the generating process, we utilize inverse loss $\mathcal{L}_{inv}$ to enforce the one-to-one mapping between the prior information extracted by $E_P^R$ and domain information of $R(I_i)$. The inverse loss is expressed as

$$\mathcal{L}_{inv} = ||E_P^R(I(i)) - E_D^R(R(I_i))||_2 + ||E_P^S(I(i)) - E_D^S(S(I_i))||_2 \tag{8}$$

**Total loss and implementation details.** We summarize all aforementioned losses and obtain the total loss.

$$\mathcal{L}_{total} = \lambda_{adv}\mathcal{L}_{adv} + \lambda_{content}\mathcal{L}_{content} + \lambda_{cf}\mathcal{L}_{cf} + \lambda_{phy}\mathcal{L}_{phy} + \lambda_{recon}\mathcal{L}_{recon} + \lambda_{inv}\mathcal{L}_{inv} \tag{9}$$

where the hyper-parameters $\lambda_{adv}$, $\lambda_{content}$, $\lambda_{cf}$, $\lambda_{phy}$, $\lambda_{recon}$ and $\lambda_{inv}$ control the importance of each term. We use the compound total loss as the final objective to train our model. The loss weights are set to $\lambda_{adv} = 2$, $\lambda_{content} = 130$, $\lambda_{cf} = 100$, $\lambda_{phy} = 1$, $\lambda_{recon} = 200$, $\lambda_{inv} = 500$, The content encoders $E_C$ consist of several strided convolutional layers followed

by residual blocks [60]. The domain information encoders $E_D$ and prior information encoders $E_P$ include several strided convolution layers followed by a global pooling layer and a fully connection layer. The decoder $G$ is constructed with two branches. One branch takes the content code as input, including several residual blocks. The other branch takes the domain code as input, and processes it using a multilayer perceptron (MLP) [61] to produce a set of affine parameters $\gamma$ and $\beta$. Then, the content code and domain code are combined by adaptive instance normalization (AdaIN) [62] layers.

$$AdaIN(a, \gamma, \beta) = \gamma(\frac{a - \mu(a)}{\sigma(a)}) + \beta \qquad (10)$$

where $a$ is the activation of the previous convolutional layer in branch one, $\mu$ and $\sigma$ are channel-wise and standard deviation, respectively. And finally decode the transferred image with an upsampling module consisting of several deconvolutional layers.

Figure 3 shows some typical results of our IDGAN. The intrinsic decomposition is only used in the training process as data augmentation method, so it does not effect the real-time of the detection process.

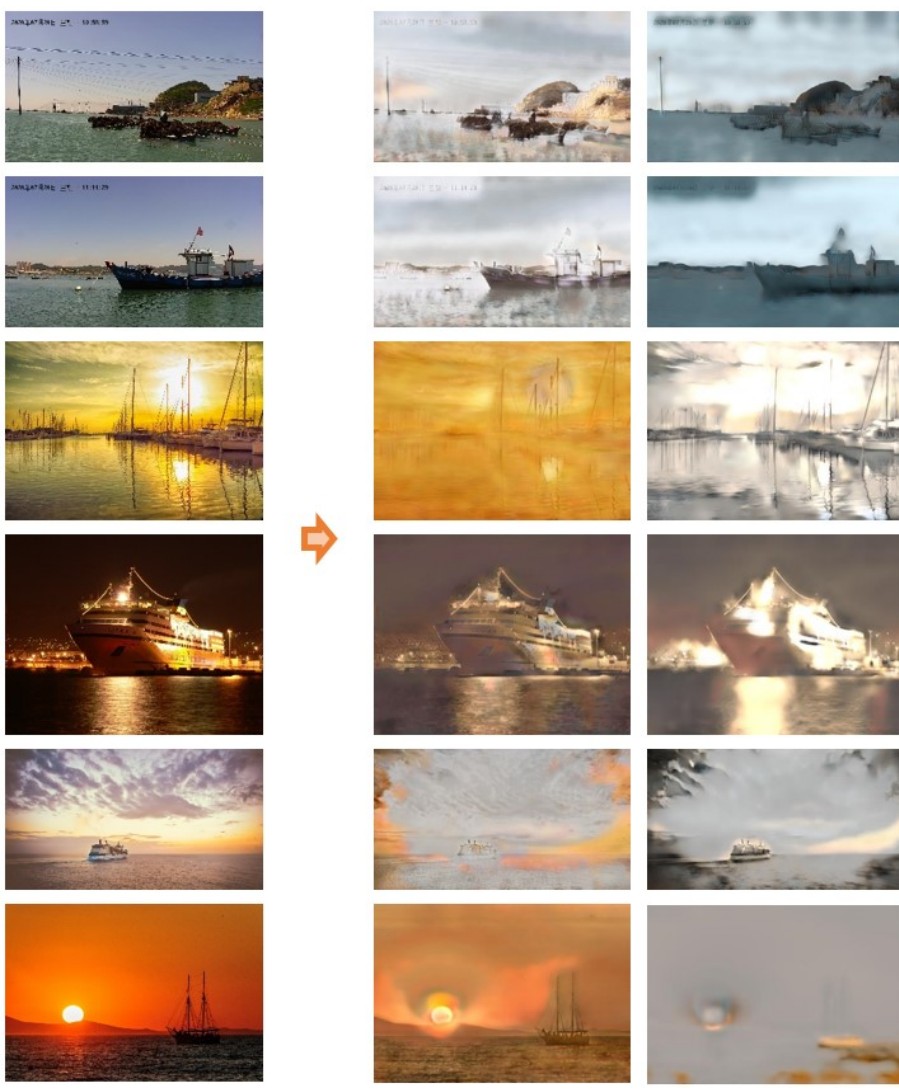

**Figure 3.** Examples of image intrinsic decomposition.

### 3.3. Multi-Scale Feature Fusion Object Detection Network

The architecture of MFFDet can be divided into three parts: Backbone, Neck, and Head, as shown in Figure 4.

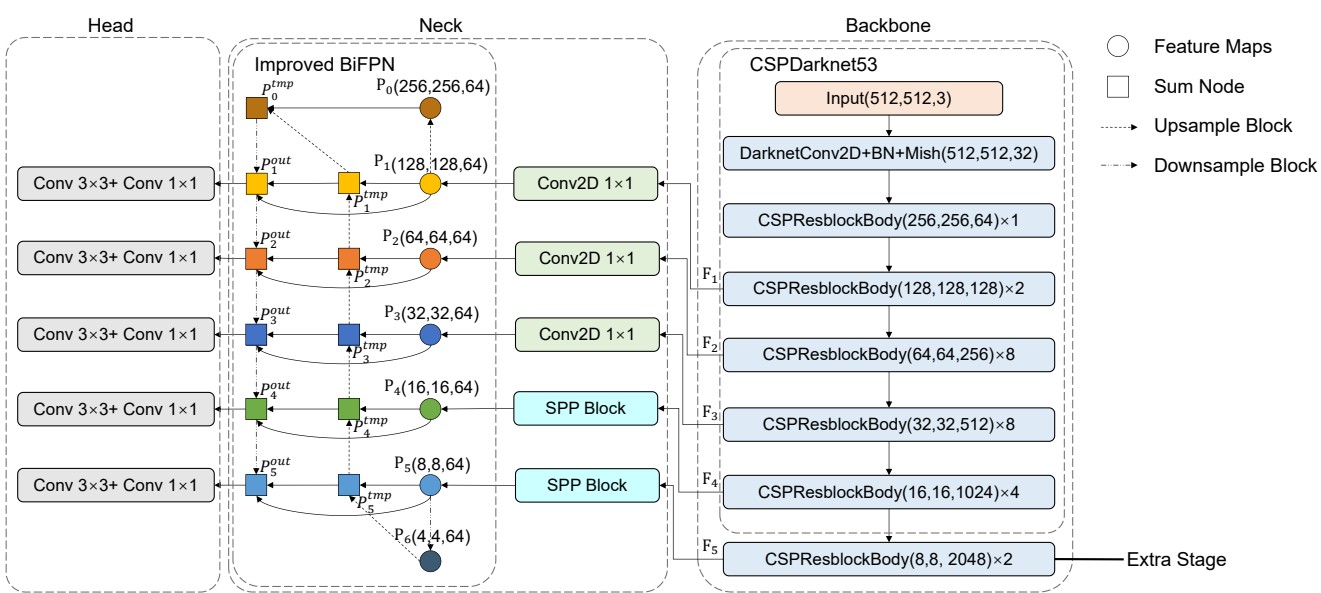

**Figure 4.** Architecture of MFFDet.

### 3.3.1. Backbone

The backbone part extracts the semantic information of the image by stacking convolutional layers. The receptive field expands with the increase in layer depth. We use a deeper CSPDarknet53 [6] as the backbone, which extends CSPDarknet53 with an extra stage, to obtain a larger receptive field and more available semantic information, as shown in Figure 4. The extra stage is composed of 2 CSPResblockBodies. Every stage in the backbone part changes the resolution of the feature maps to half. And we feed the output of the last five stages to the neck part for further feature fusion process.

### 3.3.2. Neck

The neck part further processes the features extracted by the backbone part and generates the suitable features for the head part to obtain detection result.

The SPP block [24] can fuse features of different resolution, which enhance the robustness of the network for scale variant. In detail, the SPP block applies pooling layers of different sizes to process the received feature and then concatenate them to generate fused features. To better fuse features, we set 2 SPPs after the $F_4$ and $F_5$ layers of the backbone part. The pooling size of the SPPs are set as 5, 9 and 13.

The feature maps from different layers of the backbone part are of different sizes, meaning that these feature maps obtain different resolutions and are sensitive to objects of different scales. A common way to fuse features with different resolutions is to adjust the feature map sizes to the same and then simply add or concatenate them. BiFPN [63] is an effective feature pyramid network that utilizes attention mechanism and bidirectional fusion. However, in the process of BiFPN, it removed the fusion nodes of the deepest and the shallowest layers, for there exist no deeper or shallower features, as shown in Figure 5a. But the features of these layers are important for multi-scale detection, especially for the detection of objects with extreme scales (extremely small or extremely large). So, we generate two auxiliary feature maps $P_0$ and $P_6$ by Upsample Block and Downsample Block to complete the fusion process, as shown in Figure 5b.

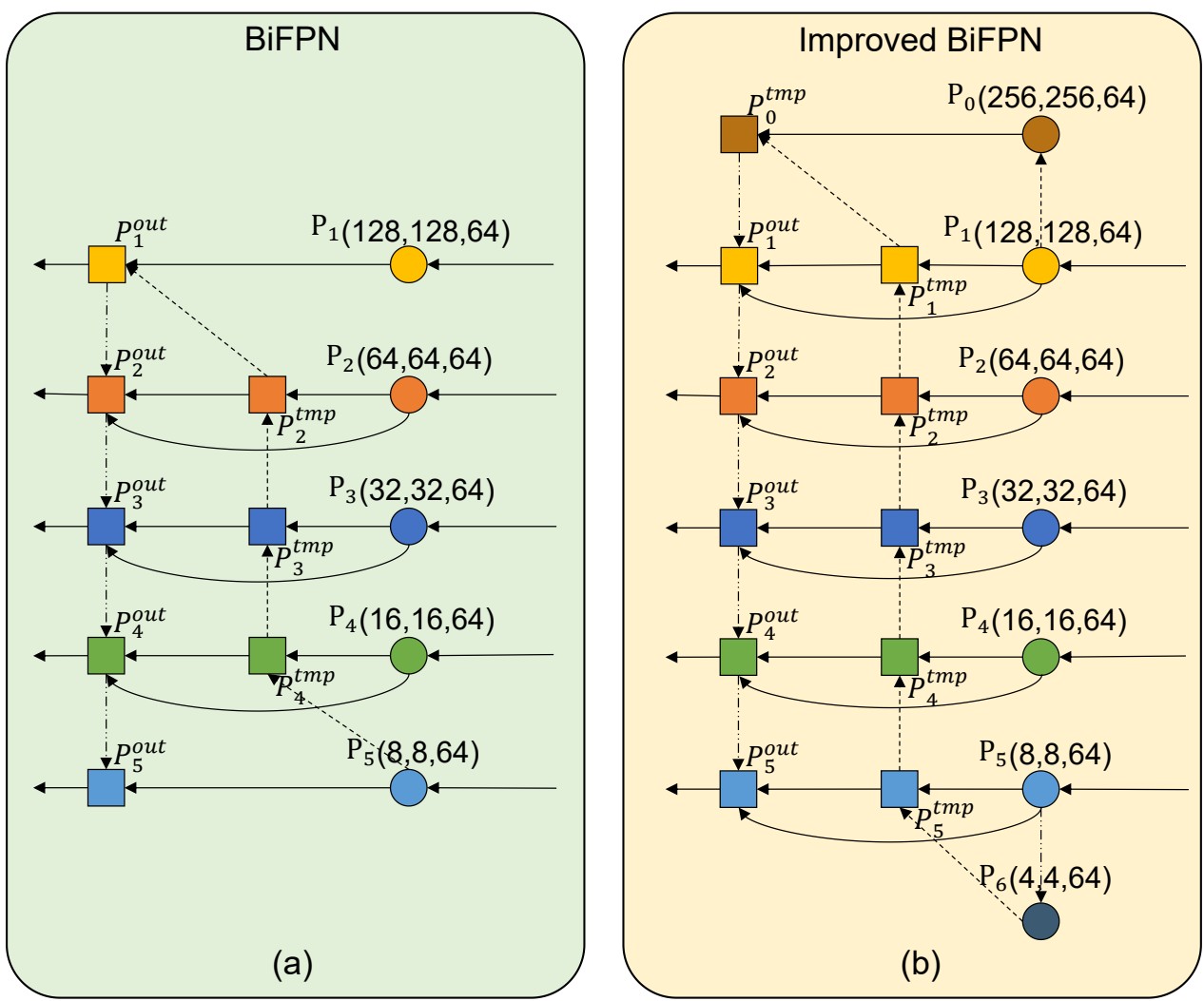

**Figure 5.** Structural comparison between BiFPN and Improved BiFPN. (**a**) BiFPN. (**b**) Improved BiFPN.

The auxiliary feature maps are not directly used to predict objects, but can enhance the fusion process of features of different resolutions, especially the ones with the biggest and the smallest scales. In detail, the Improved BiFPN utilize Upsample Blocks and Downsample Blocks to adjust the sizes of features and fuse features of different resolution. The Downsample Block utilized strided convolutional layers to decrease the sizes of the feature maps and the Upsample Block utilized strided deconvolutional layers to increase the sizes of the feature maps.

### 3.3.3. Head

The head part predicts the final results by processing the features generated by the neck part. In this part, we use a YOLO-style anchor-based head. Assume that the feature generated from the neck part is of the shape of $H \times W \times C$, where $H$, $W$, and $C$ refer to height, width and channel. The prediction result shape is $H \times W \times [A \times (4 + 1 + CLS)]$, where $A$ is the prior anchors per position, 4 is the bounding box offsets, 1 is objectness prediction, and $CLS$ is the amount of categories. Moreover, we select five scales of feature maps with different scales for prediction to obtain better multi-scale detection performance. The selected feature maps are of shapes of $8 \times 8$, $16 \times 16$, $32 \times 32$, $64 \times 64$ and $128 \times 128$. The predicted boxes need to be further processed by non-maximum suppression (NMS) [64]

to obtain the final results. And we use the same loss function as yolov4 [6] to optimize our model.

*3.4. Weighted-SWA*

To further improve the generalization capability of our method, we propose an improved Weighted-SWA. SWA [65] is an effective neural network ensembling method that can accelerate the training process and improve the generalization of the network. In this section, we propose Weighted-SWA, which use an entropy value method to weight the checkpoints, instead of simply averaging them.

Entropy represents the chaos of a system. The information entropy can be used as the objective weight while evaluating a subject, and it can also be used to objectively assign weights to multiple subjects. By estimating the degree of change of each subject, we can obtain the degree of contribution of the subject, and the weight of the subject with greater change is also greater. Assume the index matrix $X = (x_{ij})_{m \times n}$, where $m$ is the amount of subjects and $n$ is the amount of evaluation indicators. For an indicator $x_j$, the more discrete the index matrix is, the greater the weight of the index in the evaluation is.

The steps for Weighted SWA are as follows:

1. **Performance evaluation of checkpoint.** When the loss function basically does not show a decreasing trend during the training process, it continues to be trained for an additional period of time using the cyclic learning rate. Then, additional $m$ checkpoint models are obtained and evaluated on the dataset to obtain their performance on the $n$ categories.

2. **Standardization of data for every indicator.** The indicators used in the index matrix usually include positive and negative indicators. But there exists no negative indicator in the performance of the models, for what only positive indicators are used. To standardize the indicators, $x_{ij}^* = \frac{x_{ij} - x_{min}}{x_{max} - x_{min}}$. Then, use Z-score to obtain the proportion of model $i$ in indicator $j$, $Z_{ij} = \frac{x_{ij} - \bar{x}_j}{s_j}$, where $s_j$ is the standard deviation $s_j = \sqrt{\frac{1}{m}[(x_{1j} - \bar{x}_j)^2 + (x_{2j} - \bar{x}_j)^2 + \cdots + (x_{mj} - \bar{x}_j)^2]}$.

3. **Calculate the entropy and entropy redundancy.** The information entropy of indicator $j$ is $e_j = -K \sum_{i=1}^{m} Z_{ij} \ln Z_{ij}$, where $K$ is a positive number. The entropy maximizes when a system is completely disordered. At this point $Z_{ij}$ for the given $j$ all the same and $Z_{ij} = \frac{1}{m}$. Here, $e_j$ takes a great value, i.e., $e_j = -K \sum_{i=1}^{m} \frac{1}{m} \ln \frac{1}{m} = K \ln m$. The entropy redundancy of indicator $j$ is $d_j = 1 - e_j$, representing the effectiveness of the indicator.

4. **Calculate the weight of indicators and the comprehensive evaluation of the models.** The greater the entropy redundancy of a certain indicator, the greater its importance for evaluation. The weight of indicator $j$ is $w_j = \frac{d_j}{\sum_{j=1}^{n} d_j}$. And the comprehensive evaluation of model $i$ is $f_i = \sum_{j=1}^{n} w_j x_{ij}^*$.

5. **Get the final model.** The internal parameters of the $m$ checkpoint models are weighted by the comprehensive evaluation $f$ and synthesized according to the integrated evaluation value to determine the optimal model.

## 4. Experiment

To demonstrate the effectiveness of our method, We conduct extensive experiments on a water surface object detection dataset. And we further test our method on an USV in the sailing process to prove the practicality.

*4.1. Dataset Preparation*

Due to the lack of a water surface dataset with harsh illumination conditions, we selected 433 images from WSODD [66] in different water scenes, 127 images from the boat-types recognition dataset, and 417 images containing common objects on the water surface from publicly available photos on the Internet to build the validation dataset. A

dataset of 977 images was formed for testing and validating the algorithm. Figure 6 shows some typical samples of this dataset.

Six common objects on the water surface were selected to be included for recognition: boat, ship, ball, bridge, harbor, and animal. Table 1 lists the number of images and instances in every category in the dataset. The object categories in this dataset are also divided relatively broadly. For example, the boat category includes rubber boats, canoes, sailboats, etc.

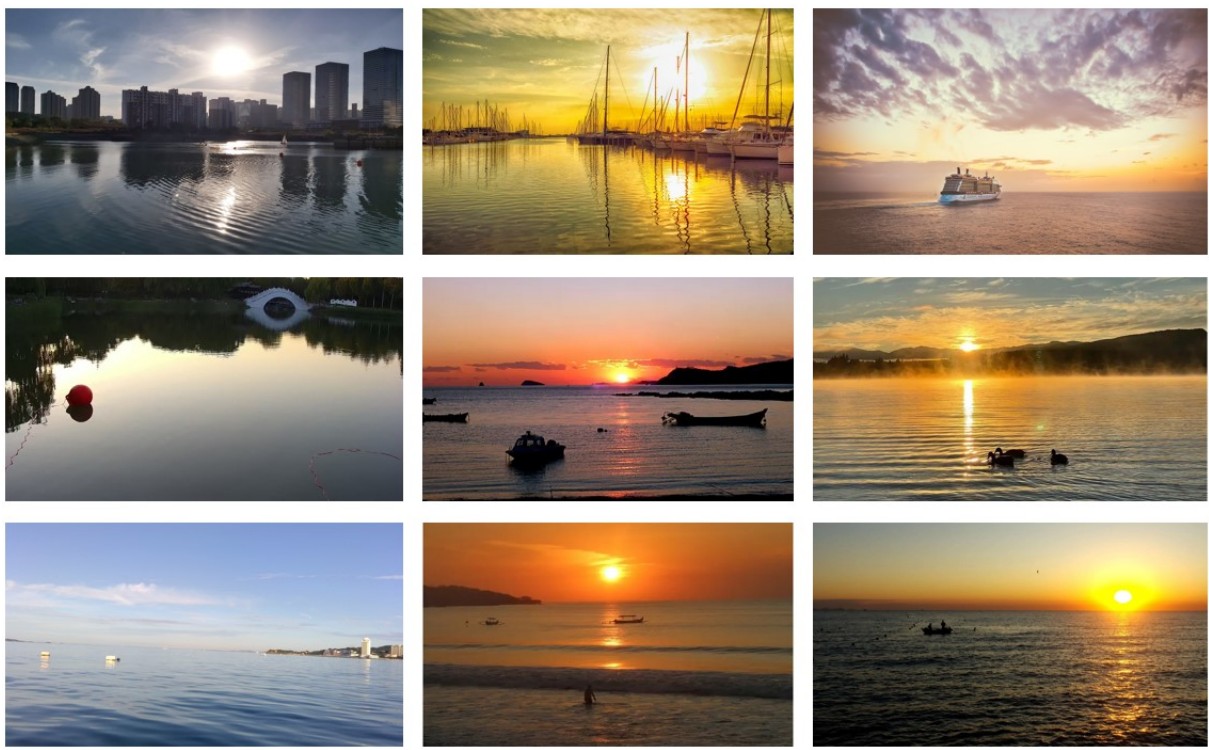

**Figure 6.** Typical samples in water surface dataset with harsh illumination conditions.

**Table 1.** The number of images and instances within every category in the dataset.

| Category | Images | Instances |
|---|---|---|
| boat | 317 | 677 |
| ship | 467 | 868 |
| ball | 170 | 210 |
| bridge | 69 | 69 |
| harbor | 70 | 77 |
| animal | 198 | 237 |
| total | 977 | 2138 |

The dataset is annotated in PASCAL VOC style, saving as xml files.

### 4.2. Experiment on Water Surface Object Detection Dataset

We compare our method with 10 deep-learning-based detection methods on the water surface object detection dataset. The compared methods include 8 effective general one-stage detectors (SSD, RetinaNet, Yolov3, RFBNet [67], M2Det [68], CenterNet, EfficientDet, and Yolov4) and 2 specialized detectors designed for water surface object detection. The operating system of the experimental platform is Ubuntu 16.04, and the GPU used is Nvidia Titan-RTX with 24 GB of RAM. We use average precision (mAP) to measure the accuracy of the detection accuracy and the IoU threshold is set at 0.5. The detection speed is measured by frames per second (FPS).

Table 2 shows the results of the baseline tests of the 11 algorithms. Columns 2 and 3 represent the FPS and mAP of different algorithms on the dataset, respectively. And columns 4–9 show the experimental results of each algorithm for every category in the dataset.

**Table 2.** Performance of 11 detection algorithms on a water surface object detection dataset.

| Method | FPS | mAP | $AP_{50}$ | | | | | |
|--------|-----|-----|------|------|------|--------|--------|--------|
| | | | **Boat** | **Ship** | **Ball** | **Bridge** | **Harbor** | **Animal** |
| SSD | 43.44 | 29.5% | 18% | 47% | 14% | 32% | 39% | 27% |
| RetinaNet | 34.22 | 23.7% | 11% | 30% | 18% | 17% | 47% | 19% |
| Yolov3 | 45.81 | 31.0% | 17% | 35% | 21% | 35% | 55% | 23% |
| RFBNet | 44.97 | 25.7% | 12% | 36% | 17% | 21% | 46% | 22% |
| M2Det | 41.11 | 29.2% | 13% | 45% | 22% | 24% | 44% | 27% |
| CenterNet | 44.09 | 31.0% | 19% | 45% | 31% | 20% | 45% | 26% |
| EfficientDet | 29.11 | 25.7% | 15% | 38% | 15% | 21% | 45% | 20% |
| Yolov4 | 46.07 | 31.8% | 17% | 37% | 21% | 33% | 59% | 24% |
| Yolov3-2SMA | 50.19 | 35.8% | 13% | 45% | 21% | 37% | **72%** | 27% |
| ShipYolo | 50.09 | 29.8% | 10% | 41% | 17% | 29% | 54% | 28% |
| **MFFDET-IDGAN** | 44.11 | **46.0%** | **27%** | **67%** | **41%** | **44%** | 69% | **28%** |

As can be seen from the table, MFFDet-IDGAN achieved the highest mAP of 46.0% on the selected dataset compared to other algorithms. There is a 10.2% improvement over Yolov3-2SMA and 22.3% higher than RetinaNet. The deeper backbone and the auxiliary feature map cause increases in calculating the detection network. Assume the input is a RGB image with a width and height of 512. The parameter quantity of our model is 162*M* and the floating point operations (FLOPs) is 86 G. Compared to YOLOv4 with 64 M parameters and 60 G FLOPs, our model can reach a 44% increase in mAP with a 30% increase in calculation. From the perspective of detection speed, our method can reach 44.11 frames per second, which is sufficient to support real-time detection. The results show that our method can effectively improve the accuracy and keep fine rapidness. By analyzing the performance of all the categories, it can be seen that our method surpasses other detectors in most categories, especially for boat and ball. In the evaluation dataset, the scales of boat and ball categories are relatively small, which is challenging for the detectors, so the significant improvement on the two categories can prove the superior multi-scale detection performance of our method. However, the animal category includes different species such as ducks and geese on the water, and the category features are more complex. And most of this category in the dataset is made up of small targets, and the semantic features are not obvious, so the improved algorithm has a limited effect in detecting them. Furthermore, for the harsh illumination condition of the dataset, the overall improvement can demonstrate the illumination adaptive property of the proposed MFFDet-IDGAN.

*4.3. Ablation Studies*

In order to verify the effectiveness and generalization performance of the intrinsic decomposition data augmentation, we choose to apply it on 4 object detection methods (Yolov3, CenterNet, Yolov4, and Yolov3-2SMA), comparing the detection performance between cutmix and IDGAN. The results are shown in Table 3. The results show that our IDGAN significantly improves the performance of the chosen algorithms. The mAP improvement can reach at most 7.6% while applied on YOLOv3 and at least 5.5% on the YOLOv3-2SMA.

**Table 3.** Validation of intrinsic image decomposition data augmentation.

| Method | Cutmix | | IDGAN | |
|---|---|---|---|---|
| | FPS | mAP | FPS | mAP |
| Yolov3 | 45.81 | 31.0% | 45.87 | 38.6% |
| CenterNet | 44.09 | 31.0% | 44.30 | 37.1% |
| Yolov4 | 46.07 | 31.8% | 46.88 | 37.8% |
| Yolov3-2SMA | 50.19 | 35.8% | 49.87 | 41.3% |

And we also verify the effectiveness of Weighted-SWA on the chosen detectors, by comparing the performance of original detector, with SWA and with Weighted-SWA. The results are shown in Table 4. The results show that the Weighted-SWA can obtain more significant improvement.

**Table 4.** Validation of Weighted-SWA.

| Method | Original Algorithm mAP | SWA mAP | Weighted-SWA mAP |
|---|---|---|---|
| Yolov3 | 31.0% | 32.3% | 33.8% |
| CenterNet | 31.0% | 31.2% | 31.9% |
| Yolov4 | 31.8% | 32.9% | 34.0% |
| Yolov3-2SMA | 35.8% | 36.2% | 37.2% |

To further explore the contribution of each key component of MFFDet-IDGAN to the detector, another 10 models are constructed and evaluated. The basic model is a detector with a deeper CSPDarknet53 backbone, and a YOLO-style head, without a feature fusion module. The results are shown in Table 5. The results show that any one of the components can improve the mAP of detection.

**Table 5.** Performance of each key part of MFFDet-IDGAN.

| | | | | | | | |
|---|---|---|---|---|---|---|---|
| +2 SPP | - | ✓ | ✓ | ✓ | ✓ | ✓ | ✓ |
| + Improced BiFPN | - | - | ✓ | ✓ | ✓ | ✓ | ✓ |
| + IDGAN | - | - | - | - | ✓ | ✓ | ✓ |
| + Weighted-SWA | - | - | - | - | - | ✓ | ✓ |
| Backbone→CSPDarknet53 | - | - | - | - | - | - | ✓ |
| mAP | 24.9% | 27.6% | 33.8% | 35.7% | 43.1% | **46.0%** | 42.8% |

*4.4. Practical Experiment on USV*

To further validate the practical effectiveness of the proposed algorithm, it is applied on USV180 for visual perception for verification. Figure 7 shows the USV180 and the Nvidia Jetson TX2 loaded on it for graphics processing.

In the experiment process, we recorded several videos during navigation, and implement 4 object detection methods (Yolov3, CenterNet, Yolov4 and Yolov3-SMA) on them for comparison. Here, we choose 3 representative videos for analysis, in which there exists small scale objects and strong reflection on the water surface.

Video 1 is 24 seconds long and was captured at Linghai Campus of Dalian Maritime University. The lighting condition in video 1 is fine, but the scale of objects is small, which is difficult for detection, for the long capture distance. Figure 8 shows the detection effect of MFFDet-IDGAN on some typical frames, and the small scale objects can be well-detected. Table 6 shows the detection effect of the chosen detectors. It can be seen that our MFFDet-IDGAN obtains the best performance with mAP of 91.1%.

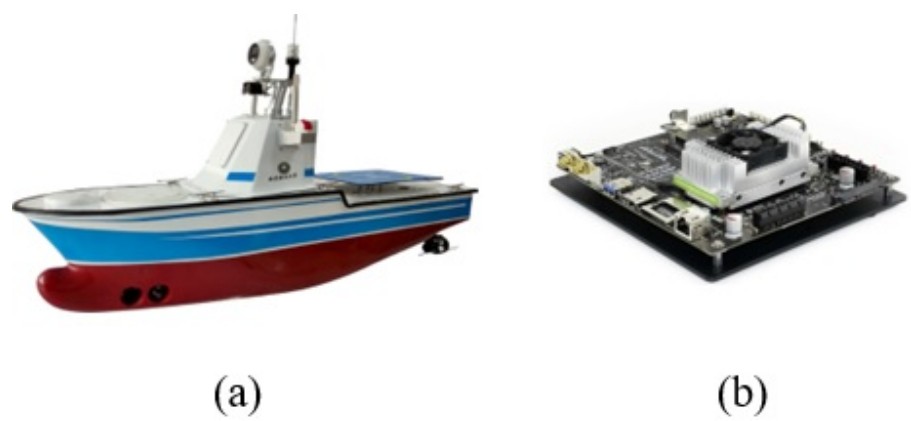

**Figure 7.** Validation platform for practical experiment. (**a**) USV180. (**b**) USV sensing platform NVIDIA Jetson TX2.

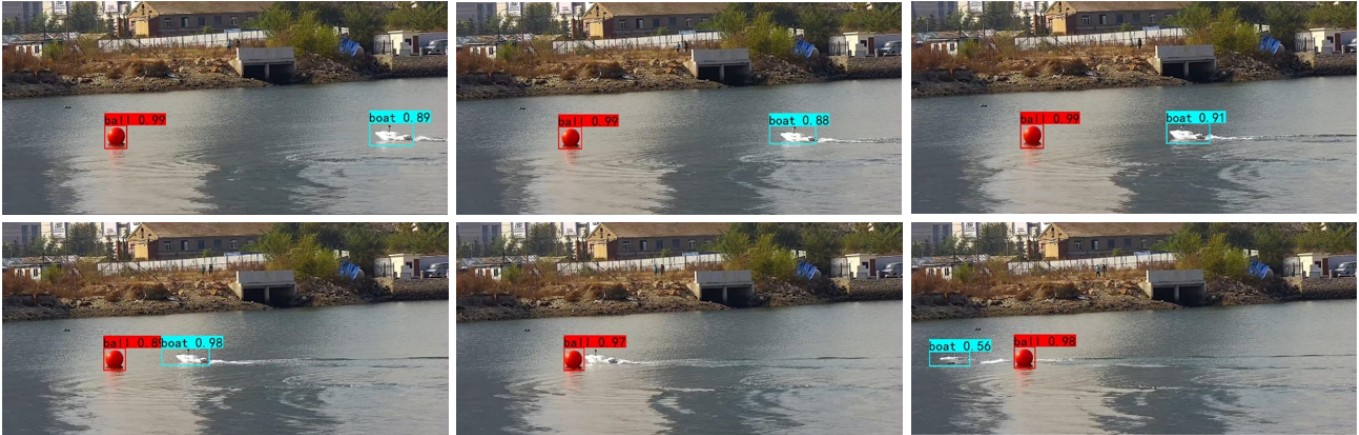

**Figure 8.** Detection effect of typical frames in video 1.

**Table 6.** Detection effect in video 1.

| Method | Total Frames | Valid Frames | mAP | FPS |
|:---:|:---:|:---:|:---:|:---:|
| Yolov3 | 600 | 587 | 88.6% | 8.58 |
| CenterNet | 600 | 587 | 79.6% | 8.91 |
| Yolov4 | 600 | 587 | 90.9% | 9.22 |
| Yolov3-2SMA | 600 | 587 | 87.2% | 10.03 |
| **MFFDet-IDGAN** | 600 | 587 | **91.1**% | 8.87 |

Video 2 is 68 seconds long and was captured at Linghai Campus of Dalian Maritime University. A total of 1450 valid frames are included in 58 seconds of this video. In video 2, the scale of objects is also small, and there exists a strong reflection on the water surface. Figure 9 shows the detection effect of MFFDet-IDGAN on some typical frames. Table 7 shows the detection results of the 5 algorithms on video 2. Our method performs the best in the experiment, with mAP of 94.5%, 4.3% higher Yolov3, 3.5% higher Yolov4 and 4.9% higher than Yolov3-2SMA.

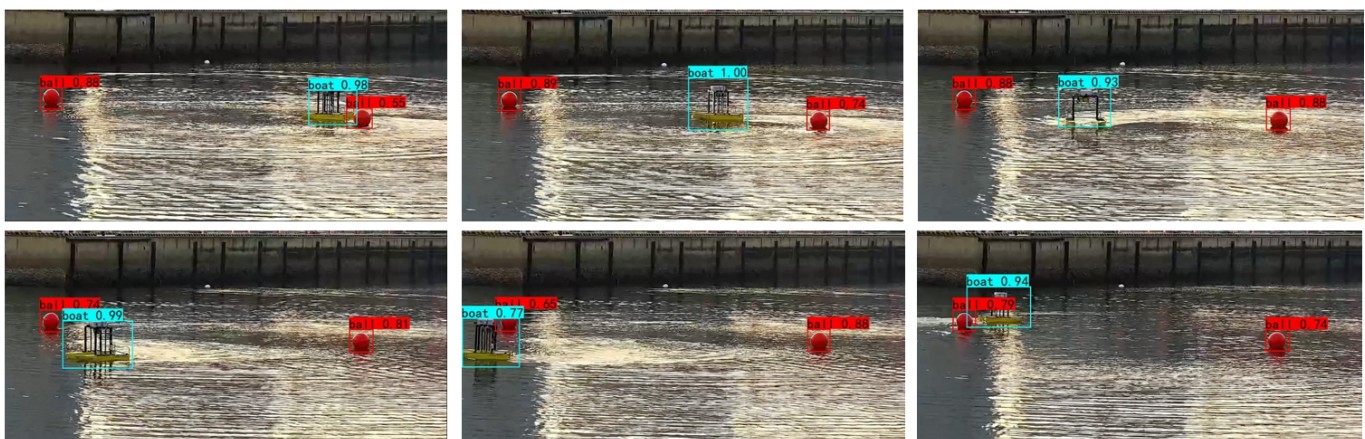

**Figure 9.** Detection effect of typical frames in video 2.

**Table 7.** Detection effect in video 2.

| Method | Total Frames | Valid Frames | mAP | FPS |
|---|---|---|---|---|
| Yolov3 | 1700 | 1450 | 90.2% | 8.44 |
| CenterNet | 1700 | 1450 | 87.9% | 8.95 |
| Yolov4 | 1700 | 1450 | 91.0% | 9.29 |
| Yolov3-2SMA | 1700 | 1450 | 89.6% | 9.97 |
| **MFFDet-IDGAN** | 1700 | 1450 | **94.5**% | 8.80 |

Video 3 is 29 seconds long and was captured by USV180 at the North Lake of Liangxiang Campus of Beijing Institute of Technology. The video records the process of another USV driving away from USV180 and there is a strong reflection on the water surface. All frames of this video are valid. Figure 10 shows the effect ofIn future works, we plan to make the detector lightweight to achieve real time detection for the resource-limited USVs on the detection of some typical frames. Table 8 shows the detection results of the 5 algorithms on video 3. As can be seen from the table, in terms of detection accuracy, compared to the other 4 algorithms, MFFDet-IDGAN improves by 4.1% over Yolov4, 6.5% over Yolov3, 7.0% over Yolov3-2SMA, and 10.0% over CenterNet.

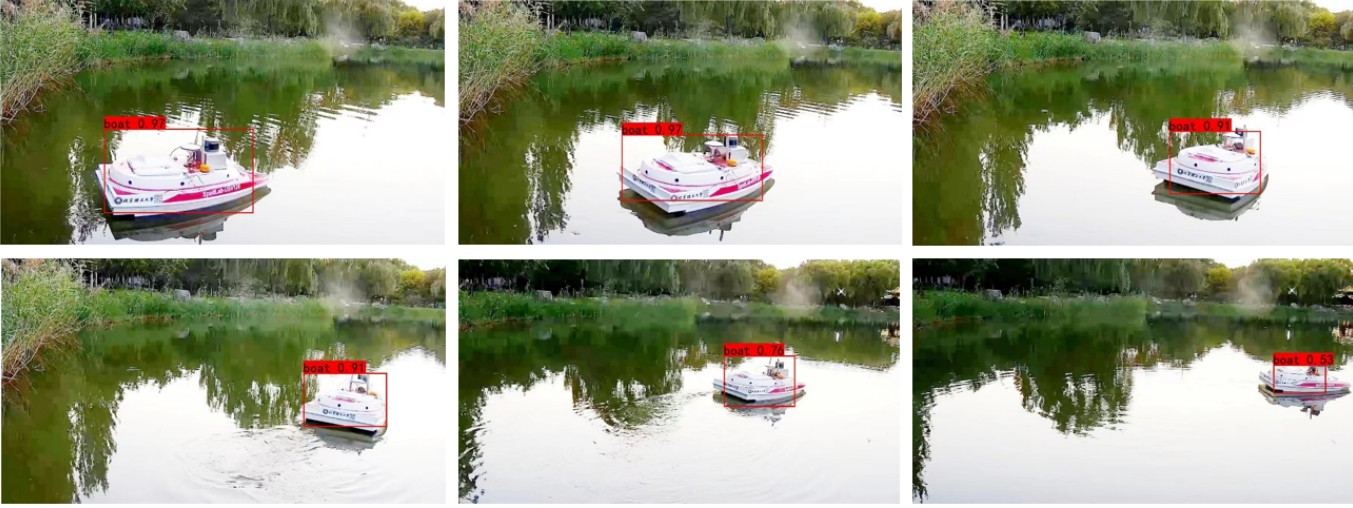

**Figure 10.** Detection effect of typical frames in video 3.

**Table 8.** Detection effect in video 3.

| Method | Total Frames | Valid Frames | mAP | FPS |
|---|---|---|---|---|
| Yolov3 | 725 | 725 | 89.7% | 8.48 |
| CenterNet | 725 | 725 | 86.2% | 9.01 |
| Yolov4 | 725 | 725 | 92.1% | 9.27 |
| Yolov3-2SMA | 725 | 725 | 89.2% | 9.93 |
| **MFFDet-IDGAN** | 725 | 725 | **96.2**% | 8.82 |

In all 3 videos, MFFDet-IDGAN obtains the best performance of the 5 methods, which demonstrates our method of practical value. Moreover, in video 2 and 3, the improvement is more significant than that in video 1, for the strong reflection on the water surface harms the accuracy of other detectors, which further proves the effectiveness of IDGAN to reduce the effect of harsh illumination.

## 5. Discussion

According to the results of the experiments, the intrinsic decomposition data augmentation can deal well with complex illumination. However, it will triple the number of training samples along with triple the time cost, which is a serious issue when using a large dataset. By observation of experiments, we found that the decomposed reflectance image of those extremely bright source images tend to be a completely white image, which we think will provide little benefit or even be harmful for the detection network. Considering this situation, we think it a feasible method to evaluate the contribution of the decomposed samples, and remove the samples that contribute little for the network to accelerate the training process.

In addition, besides the challenges we analyze in the introduction, we met many other problems with detecting while sailing, such as the interference of mist and the fake target caused by the reflection of the water surface. So, more studies need to be made to achieve more accurate and more robust water surface object detection.

## 6. Conclusions

In this paper, we analyze the difficulty of water surface object detection, especially on the scenes with harsh illumination conditions. Accordingly, we propose MFFDet-IDGAN to address these problems. The intrinsic decomposition method is introduced as a data augmentation method, which decomposes image optical prior knowledge to increase the feature diversity of training samples. And a multi-scale feature fusion object detection network MFFDet is proposed to deal with the scale variant of the objects on the water surface. The network utilizes a deeper CSPDarknet53 in the backbone to extract more semantic information. We fuse the features extracted from different convolutional layers by 2 SPP blocks and Improved BiFPN. The Improved BiFPN adapts auxiliary features to obtain better fusion effectiveness. And an improved model ensembling method Weighted-SWA is proposed, which makes use of an entropy evaluation method to weight the checkpoint and obtain the final model to improve the generalization performance. The experiment on the water surface object detection dataset shows that our method achieved 44% improvements over the baseline. We further ported our method on a USV to verify its practical effectiveness, and the results show that our method can better detect the object during navigation than the comparison methods at equal speed.

**Author Contributions:** Project administration, Z.Z. and X.Z.; supervision, Z.Z.; methodology, Z.L.; software, Z.L. and J.S.; writing—original draft preparation, Z.L. and J.S.; formal analysis, J.S.; visualization, Z.L.; data curation, J.S.; writing—review and editing, L.X. and X.Z. All authors have read and agreed to the published version of the manuscript.

**Funding:** This research received no external funding.

**Institutional Review Board Statement:** Not applicable.

**Informed Consent Statement:** Not applicable.

**Data Availability Statement:** The experimental data used for verification in this paper are publicly available in Figshare with the identifier: https://doi.org/10.6084/m9.figshare.22874387.

**Conflicts of Interest:** The authors have no conflict of interest to declare that are relevant to the content of this article.

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
