# Peer review of "Illumination Adaptive Multi-Scale Water Surface Object Detection with Intrinsic Decomposition Augmentation"

_jmse, doi:10.3390/jmse11081485_

Round 1

Reviewer 1 Report

This study proposes a robust water surface object detection method named multi-scale feature fusion network with intrinsic decomposition generative adversarial network data augmentation (MFFDet-IDGAN). We introduce intrinsic decomposition as data augmentation for object detection to achieve illumination adaptation. And an intrinsic decomposition generative adversarial network (IDGAN) is proposed to achieve unsupervised intrinsic decomposition. Moreover, the multi-scale feature fusion network (MFFDet) adopts an improved bidirectional feature pyramid network (BiFPN) and spatial pyramid pooling (SPP) blocks to fuse features of different resolutions for better multi-scale detection. And an improved weighted stochastic weight averaging (SWA) is proposed and applied in the training process to improve the generalization performance. Our method is tested on the Water Surface Object Detection Dataset (WSODD) and a  real USV in the sailing process. Both of the results show that our method can surpass the comparison detectors in detection accuracy with equal rapidness. I read the article, and this paper is well-written and contributes to the body of literature. However, I do have some concerns with the current version that need correction during the revision. I believe the following comments can further enhance this paper's quality.

1-      In the abstract, please add the performance improvements with numbers and %ages rather than fuzzy words.

2-      In the introduction, please write the implications (theoretical and technical significance) of your work in the contribution section by highlighting existing problems well. Some contents are there 32~37, but not very convincing.

3-      The paper organization should be included in the introduction section.

4-      Please write the conclusion as a separate section by summarizing the main contents.

5-      Also, the authors should provide some analysis regarding the time and space complexity of the proposed method.

6-      Some more discussion about the result before the conclusion is desirable. Authors can provide some details of experiments and challenges that can stem from working in dynamic environments.

7-      Although this method seems rational, what are some challenges or limitations of this work?

8-      What is the uniqueness of this paper compared to existing work? I think many such works have already been proposed with extensive evaluations.

9-      Do authors apply any optimization in data augmentation?

10-  The title can also be improved by adding the method information as well.

11-  Symbols can be written in math mode.

12-  Some more papers can be included in the introduction section.

English is fine in most parts.

Author Response

Dear reviewer,

We are grateful for your careful and professional comments to our paper. Each of your suggestions is indeed helpful for us to improve our work, and we have made modifications to address the drawbacks in our manuscript. The revised manuscript is in the attachment and the modified contents are highlighted in red followed by description in blue and wrapped in parenthesis. We made point-to-point response for your comments in the following part.

Point 1:

Comment: In the abstract, please add the performance improvements with numbers and %ages rather than fuzzy words.

Reply: We have replaced the fuzzy words with quantitative performance improvement in the abstract to give a clearer statement to the readers.

Point 2:

Comment: In the introduction, please write the implications (theoretical and technical significance) of your work in the contribution section by highlighting existing problems well. Some contents are there 32~37, but not very convincing.

Reply: To better illustrate the implications of our work, we have added some analysis about the existing problems in the background section and given a better corresponding between the existing problems and our solutions in the contribution section.

Point 3:

Comment:The paper organization should be included in the introduction section.

Reply: We have reorganized the Introduction section and added organization of the paper to it.

Point 4:

Comment: Please write the conclusion as a separate section by summarizing the main contents.

Reply: We have written the conclusion to summarize our main work as a separate section.

Point 5:

Comment: Also, the authors should provide some analysis regarding the time and space complexity of the proposed method.

Reply: We have added analysis about time and space complexity in the experiment section and give the parameter quantity and FLOPs of our model.

Point 6:

Comment: Some more discussion about the result before the conclusion is desirable. Authors can provide some details of experiments and challenges that can stem from working in dynamic environments.

Reply: We have added some analysis in the experiment section, and we give some details and challenges in the discussion section.

Point 7:

Comment: Although this method seems rational, what are some challenges or limitations of this work?

Reply: Thank you for pointing out an important question. According to the results of the experiments, our method can increase the detection effect especially in harsh illumination scenes. However, our data augmentation method will triple the number of the training samples along with triple time cost, which is a serious issue when using a large dataset. Considering this drawback, we are trying to find a method to evaluate the contribution of the decomposed samples, and remove the samples that contribute little for the network to accelerate the training process. And we have added this issue to the discussion part to give the readers a possible optimization direction.

Point 8:

Comment: What is the uniqueness of this paper compared to existing work? I think many such works have already been proposed with extensive evaluations.

Reply: We are sorry that we didn’t give enough analysis about existing works and make you confuse. While investigating and surveying on existing works, we found that most works about water surface object detection pay no attention to the harsh illumination condition on water surface scenes, while the works that take steps to deal with the illumination condition only give a rough preprocess to the input image. With the purpose of addressing this issue without introducing extra calculating, we propose to utilize intrinsic decomposition as data augmentation to give more prior knowledge to the detection network. To our best knowledge, we are the first to take intrinsic decomposition as data augmentation to make the model illumination adaptive, which conduct the main novelty of our work. And to get an effective detector, we improve the baseline to get better multi-scale detection performance and utilize a new model ensembling method to get better generalization. To better illustrate this to the readers, we have supplemented the introduction section with more analysis.

Point 9:

Comment: Do authors apply any optimization in data augmentation?

Reply: In the ablation study, to verify the effectiveness of our intrinsic decomposition data augmentation method, we didn’t do any optimization. But in the implementation process, we additionally utilize Mosaic data augmentation to get better performance.

Point 10:

Comment: The title can also be improved by adding the method information as well.

Reply: We have add our core method to the title to give the readers more intuitive information.

Point 11:

Comment: Symbols can be written in math mode.

Reply: We have changed the symbols to math mode.

Point 12:

Comment: Some more papers can be included in the introduction section.

Reply: While supplementing the introduction section, we have added more reference papers there.

Quality of English:

Reply: We have re-examined our paper to check and correct other existing mistakes.

Thanks again for your constructive opinions for our paper. Your comments are of great guiding significance for our paper and will help us in the future works.

Best regards,

Zhiguo Zhou

Reviewer 2 Report

In general, the scientific approach of the paper is valid and the results are rather encouraging.

Although harsh illumination conditions were the baseline for testing it is curious, how does the method perform in non-harsh illumination conditions. In short, can it be used as a general purpose water surface object detection method?

A crucial equation (9) is full of constants that should be fine tuned rather than arbitrarily selected. Perhaps the efficiency of the method could be further improved that way.

The algorithm is presented in a rather general way that seems more like a report of accomplishments than a self-sufficient description of the methodology. However, that is somewhat understandable, considering the number of interconnected elements that had to be presented.

Some language mistakes even in the abstract "multi-sacle"; in general, language is rough at times.

Author Response

Dear reviewer,

Thank you very much for your professional review on our work. And your comments are indeed helpful for us to re-examine our work. We have made modifications to address the drawbacks in our manuscript and the revised manuscript is in the attachment. The modified contents are highlighted in red followed by description in blue and wrapped in parenthesis. We made point-to-point response for your comments in the following part.

Point 1:

Comment: Although harsh illumination conditions were the baseline for testing it is curious, how does the method perform in non-harsh illumination conditions. In short, can it be used as a general purpose water surface object detection method?

Reply: In Section “Practical experiment on USV”, we test our method on three scenes, one of which is in fine illumination condition, and our method still outperforms other methods. This can demonstrate that our method can work well in non-harsh illumination conditions. So our method can be used as a general water surface object detection method.

Point 2:

Comment: A crucial equation (9) is full of constants that should be fine tuned rather than arbitrarily selected. Perhaps the efficiency of the method could be further improved that way.

Reply: These hyper-parameters controls the importance of each item in the network while training. This type of hyper-parameters is often determined experimentally and they are usually adjusted exponentially, so it’s hard to give a feasible range for the readers. To this end, we provide a set of parameters that are effective according to our experiment as reference.

Point 3:

Comment: The algorithm is presented in a rather general way that seems more like a report of accomplishments than a self-sufficient description of the methodology. However, that is somewhat understandable, considering the number of interconnected elements that had to be presented.

Reply: We are grateful for your kindness. In the process of building this paper, it’s a difficult problem to present our method as detailed as possible while not make it too lengthy, for our method involve a generative adversarial network, some improved modules to the object detection baseline and an improved model ensembling method. We have done our best to present our method in a way that the readers can realize without too many extra efforts.

Quality of English:

Comment: Some language mistakes even in the abstract "multi-sacle"; in general, language is rough at times.

Reply: We are very grateful for your careful review of our paper, and we have re-examined our paper to check and correct other existing mistakes.

Thanks again for your kind and professional advice.

Best regards,

Zhiguo Zhou

Round 2

Reviewer 1 Report

I have carefully read the revised work along with the author's responses. The authors have made the desired changes in the paper satisfactorily, and therefore, I do not have any further comments on this paper. I would like to take this opportunity to congratulate the authors on their nice work in the USV field.